# Is Robotic Assisted Colorectal Cancer Surgery Equivalent Compared to Laparoscopic Procedures during the Introduction of a Robotic Program? A Propensity-Score Matched Analysis

**DOI:** 10.3390/cancers14133208

**Published:** 2022-06-30

**Authors:** Peter Tschann, Markus P. Weigl, Daniel Lechner, Christa Mittelberger, Tarkan Jäger, Ricarda Gruber, Paolo N. C. Girotti, Christof Mittermair, Patrick Clemens, Christian Attenberger, Philipp Szeverinski, Thomas Brock, Jürgen Frick, Klaus Emmanuel, Ingmar Königsrainer, Jaroslav Presl

**Affiliations:** 1Department of General and Thoracic Surgery, Academic Teaching Hospital Feldkirch, Carinagasse 47, 6800 Feldkirch, Austria; markus.weigl@lkhf.at (M.P.W.); daniel.lechner@lkhf.at (D.L.); christa.mittelberger@lkhf.at (C.M.); paolo.girotti@lkhf.at (P.N.C.G.); thomas.brock@lkhf.at (T.B.); juergen.frick@lkhf.at (J.F.); ingmar.koenigsrainer@lkhf.at (I.K.); 2Department of Surgery, Paracelsus Medical University/Salzburger Landeskliniken (SALK), 5020 Salzburg, Austria; ta.jaeger@salk.at (T.J.); r.gruber@salk.at (R.G.); k.emmanuel@salk.at (K.E.); j.presl@salk.at (J.P.); 3Department of Surgery, St. John of God Hospital, Teaching Hospital of Paracelsus Medical University, 5020 Salzburg, Austria; christof.mittermair@bbsalz.at; 4Department of Radio-Oncology, Academic Teaching Hospital, 6800 Feldkirch, Austria; patrick.clemens@lkhf.at; 5Private University in the Principality of Liechtenstein, 9495 Triesen, Liechtenstein; christian-attenberger@outlook.com; 6Institute of Medical Physics, Academic Teaching Hospital Feldkirch, 6800 Feldkirch, Austria; philipp.szeverinski@lkhf.at

**Keywords:** robotic assisted surgery, laparoscopic surgery, oncological findings, quality of resection

## Abstract

**Simple Summary:**

The introduction of a robotic program is challenging and requires extensive experience in minimally invasive surgery. Short-term outcomes and oncological quality should not differ between robotic and laparoscopic surgery. To our knowledge, no data on the quality of surgery at the time of introduction of the robotic platform are available. The aim of this study was to compare short-term outcomes and oncological findings of robotic-assisted colorectal resections with those of conventional laparoscopic surgery within the first three years after the introduction of the robotic platform.

**Abstract:**

Background: Robotic surgery represents a novel approach for the treatment of colorectal cancers and has been established as an important and effective method over the last years. The aim of this work was to evaluate the effect of a robotic program on oncological findings compared to conventional laparoscopic surgery within the first three years after the introduction. Methods: All colorectal cancer patients from two centers that either received robotic-assisted or conventional laparoscopic surgery were included in a comparative study. A propensity-score-matched analysis was used to reduce confounding differences. Results: A laparoscopic resection (LR Group) was performed in 82 cases, and 93 patients were treated robotic-assisted surgery (RR Group). Patients’ characteristics did not differ between groups. In right-sided resections, an intracorporeal anastomosis was significantly more often performed in the RR Group (LR Group: 5 (26.31%) vs. RR Group: 10 (76.92%), *p* = 0.008). Operative time was shown to be significantly shorter in the LR Group (LR Group: 200 min (150–243) vs. 204 min (174–278), *p* = 0.045). Conversions to open surgery did occur more often in the LR Group (LR Group: 16 (19.51%) vs. RR Group: 5 (5.38%), *p* = 0.004). Postoperative morbidity, the number of harvested lymph nodes, quality of resection and postoperative tumor stage did not differ between groups. Conclusion: In this study, we could clearly demonstrate robotic-assisted colorectal cancer surgery as effective, feasible and safe regarding postoperative morbidity and oncological findings compared to conventional laparoscopy during the introduction of a robotic system.

## 1. Introduction

Oncological results of minimally invasive surgery for colorectal cancer are comparable with those of open colectomies [1,2]. Since laparoscopy was first introduced for colorectal diseases in the early 1990s, its popularity has increased rapidly [3]. The benefits of laparoscopy, such as less incisional trauma, faster recovery, less pain, faster intestinal passage, shorter length of hospital stay and lower rates of incisional hernia, are well-described in the past literature [4,5].

Robotic systems were designed to overcome the limitations of laparoscopy, offering three-dimensional visualization, a surgeon-controlled stable camera and a third arm for fixed retraction, increased degrees of freedom, improved articulation and finally stabilization of tremors [6]. Robotic surgery is reported to have comparable short-term outcomes [6]. Moreover, first studies reported similar oncological long-term data for both techniques [7,8]. The evolution of robotic surgery is well-illustrated by bibliometric data, as more and more studies are published in the literature [9]. While clinical outcomes generally seem to be equivalent to those achieved via laparoscopy, robotic platforms require significantly more initial costs with longer operation room times [10]. The common consensus is that robotic surgery has a shorter learning curve [10] and better controlled teaching options compared to laparoscopic techniques. However, it should be considered that at the beginning of a robotic program, only specialized surgeons are able to perform these procedures. The current literature comparing robotic, open and laparoscopic resections includes reviews, comparative studies, meta-analyses and one randomized controlled trial [6]. They could identify comparable clinical and oncological outcomes when comparing robotic to laparoscopic surgery. To our knowledge, there are no clear data existing on how robotic-assisted surgery influences the surgical outcome compared to other minimally invasive colorectal resections at the beginning of a robotic colorectal program.

The aim of this work was to evaluate how the introduction of robotic-assisted surgery influenced common surgical practice of experienced minimally invasive and colorectal surgeons within the first three years of a robotic program and if this influenced peri- and postoperative morbidity and oncological findings compared to conventional laparoscopic surgery.

## 2. Methods

In accordance with ethical review guidelines (EK-0.04-403) and after an institutional review board approval, a retrospective review of patients who underwent either laparoscopic resection (LR Group) or robotic-assisted resection (RR Group) for colorectal cancer was performed. This study was conducted at two different centers at the Academic Teaching Hospital in Feldkirch and the Department of Surgery (2019–2021, *n* = 70), Paracelsus Medical University in Salzburg (2018–2020, *n =* 105) within the first three years after the introduction of robotic-assisted surgery. In Salzburg, the robotic platform was introduced one year before Feldkirch. All performed resections were consecutively enrolled. All operations were performed by four specialized and experienced colorectal surgeons in both hospitals with at least 10 years of experience in colorectal and minimally invasive surgery. Inclusion criteria were as follows: histologically verified colorectal cancer, older than 18 years of age and laparoscopic or robotic resection. Exclusion criteria were defined as: emergency cases, abdominoperineal resections and patients with synchronous metastasis at time of cancer diagnosis. Preoperative patients’ characteristics (sex, BMI, ASA score [11], localization of tumor, preoperative CEA level, previous abdominal surgery, comorbidities and preoperative hemoglobin), intraoperative (operative time, conversion rate) and postoperative findings (complication rate according to Dindo–Clavien classification [12], time to first flatus, hemoglobin on the first postoperative day, number of harvested lymph nodes, specimen scoring [13,14,15], TNM classification [16], morbidity and mortality rate and length of hospital stay) were reviewed. The primary outcomes were operative time, conversion rate, specimen scoring, harvested lymph nodes and length of hospital stay. The secondary outcomes were postoperative morbidity and oncological staging as a possible selection bias for robotic surgery. All patients in both groups underwent standard preoperative workup, which included colonoscopy, tissue biopsy to confirm colorectal cancer and contrast-enhanced computed tomography (CT) of the chest and the abdomen. In case of middle or low rectal cancer, an endoscopic ultrasound and pelvic magnet resonance imaging (MRI) were additionally performed. Moreover, all rectal cancer cases were discussed in a multidisciplinary team discussion preoperatively. Mechanical bowel preparation was carried out as standard procedure independent of the location of the tumor in both centers. In addition, oral antibiotic prophylaxis was performed routinely at the surgical department of Salzburg. A standard antibiotic prophylaxis was given perioperatively directly before surgery in both centers. 

### 2.1. Surgical Technique

All operations were performed by specialized colorectal surgeons with extensive experience in both open and laparoscopic surgery before starting the robotic-assisted program. After proctorship and training with the robotic system, the robotic program was introduced. 

Independent of surgical technique and location of tumor, all patients received a urinary catheter preoperatively. For right-sided colorectal resection, all patients were treated in supine position and surgery was performed by a single-docking, totally robotic technique using the Da Vinci Robotic Surgical System (Intuitive Surgical System, Sunnyvale, CA, USA) X at the Academic Teaching Hospital Feldkirch or Xi at the surgical department of the Paracelsus Medical University Salzburg. The robotic cart was docked on the side of the tumor occurrence. Four 8 mm robotic trocars were placed diagonally to the operation area, lying on a linear line. In case of intracorporeal anastomosis, one 12 mm trocar was placed instead of one 8 mm trocar for the Da Vinci linear stapler system. The configuration of the trocars consisted of two left-handed instruments and one right-handed instrument. Additionally, one 12 mm Airseal^®^ trocar (Conmed, Largo Florida, FL, USA) was placed for the assistant at the patient site. For left-sided resection, a Lloyd-Davis position was commonly used for trans-anal stapling or suturing. 

In both left- and right-sided resections, a vessel-first approach followed by a medial-to-lateral dissection respecting the avascular embryological planes was performed [14,17]. In case of right-sided resections, a side–side anastomosis was carried out either intracorporeally or extracorporeally. Specimen extraction was carried out via Pfannenstil incision or via umbilical incision in case of single incision or reduced port surgery. In case of sigmoid or upper rectum resection, a circular stapled end-to-end anastomosis was performed using a 28 mm anvil. Anastomosis in middle- and low-rectal-cancer resections was also performed using a 28 mm anvil, and usually a side–end anastomosis was created. A protective loop ileostomy was created in all mid- and low-rectal-cancer cases with a primary intended anastomosis. A hand-sewn colo-anal anastomosis has found application in cases of ultra-low rectal cancer cases. 

Conversion was defined as the unexpected change from minimally invasive to open surgery. Operative time was considered as the first skin incision until the last scar was closed. Drainage tube was usually placed in case of a rectal resection. In all other procedures, a drainage was not placed routinely. In case of absence of nausea, vomiting or unusual abdominal pain, oral intake was started on the first postoperative day. Peridural anesthesia was not carried out in case of minimally invasive surgery. Urinary catheter was removed on the operation day or on the first postoperative day. 

Laparoscopic resections were performed with the multiport technique at the Paracelsus Medical University in Salzburg. In the Academic Teaching Hospital in Feldkirch, the laparoscopic operations were performed with a reduced port technique using an umbilical single-port device and a 5 mm additional trocar suprasymphyseal.

### 2.2. Histopathological Examination

The removed specimen was fixed with formalin immediately. The pathological examination included a macroscopic description of the specimen, a score on the quality of the resection [13,14] and a complete histopathological staging. Scoring was routinely performed in all rectal cancer cases and on special request in all other locations. 

### 2.3. Propensity-Score-Matched Analysis (PSM)

We performed PSM to remove the confounding factors and overcome possible patient selection bias. Logistic regression (EZR Version 1.55) was used to calculate propensity score for laparoscopic and robotic individuals matched 1:1. For all patients, the propensity score was calculated based on the following variables: sex, age, neoadjuvant therapy, tumor localization, primary tumor stage and nodal status (according to TNM staging). 

### 2.4. Statistical Analysis

All statistical analysis other than PSM was realizing using the SPSS (Version 27.0, IBM, New York, NY, USA). Continuous data are represented as mean (±SD) and were assessed by either the t-test or the Mann–Whitney U test. Categorical data are presented in absolute numbers (percent) and were assessed using the chi-square test or the exact Fisher test for small samples. Data were collected using Excel© (Microsoft, Seattle) and analyzed with java-based tools and SPSS. The development of operation time and number of performed cases is included in Figure 1. A 7-case simple moving average method was performed to create the trend line. Significance was set at a *p*-value of <0.05. 

## 3. Results

Laparoscopic resection (LR Group) proceeded in 82 cases and 93 patients who underwent robotic-assisted surgery (RR Group). A total of 46.34% in the LR Group and 35.48% in the RR Group were female patients. Age (LR Group: 64 vs. RR Group: 68, *p =* 0.438), BMI (LR Group: 25.45 kg/m^2^ vs. RR Group: 25.1 kg/m^2^, *p* = 0.654), ASA stage (*p* = 0.122), tumor location, preoperative CEA level (LR Group: 2.4 µg/L vs. RR Group: 2.1 µg/L, *p* = 0.468), comorbidities (*p* = 0.328), previous surgery rate (LR Group: 18 (21.95%) vs. RR Group: 32 (34.41%), *p* = 0.056, preoperative clinical stage (*p* = 0.611) and type of preoperative therapy in the case of rectal cancer (*p* = 0.344) did not differ between groups. Based on the PSM, we selected 63 patients who underwent laparoscopic surgery and 63 patients who underwent robotic colorectal surgery. After adjusting for background factors using a PSM, the patients’ distributions were well-balanced between both groups. Patients’ characteristics before and after the PSM are shown in Table 1.

The operative method and type of anastomosis did not differ between groups. In the case of right-sided resections, an intracorporeal anastomosis was significantly more often performed in the RR Group (LR Group: 5 (26.31%) vs. RR Group: 10 (76.92%), *p* = 0.008). A protective defunctioning stoma was more often created in the robotic group (LR Group: 16 (32%) vs. RR Group: 39 (55.7%), *p* = 0.006). Operative time was shown to be significantly shorter in the LR Group (LR Group: 200 min (150–243) vs. 204 min (174–278), *p* = 0.045). A detailed analysis of the duration of the operation showed a trend of decreasing operative time with the number of performed procedures (Figure 1). 

Time to first flatus, postoperative hemoglobin value and number and severeness of complications according to Clavien–Dindo classification [12], time to stoma reversal and duration of hospital stay did not differ between groups. Conversions to an open procedure did occur significantly more often in the LR Group (LR Group: 16 (19.51%) vs. RR Group: 5 (5.38%), *p* = 0.004). Comparable results were found in the PSM (LR Group: 13 (20.63%) vs. RR Group: 3 (4.76%), *p* = 0.006). One patient in the LR Group and two patients in the RR Group died within the postoperative course due to severe complications. Patients’ mortality was caused by nonsurgical complications (one due to severe pneumonia, one due to aspiration with pneumonia and one because of multiple embolic-induced infarctions). The type of anastomosis (either intracorporeally or extracorporeally) in right colectomies was shown not to be significant after the PSM. Operative method, perioperative and postoperative results before and after the PSM are shown in Table 2.

The postoperative local tumor stage (*p* = 0.831), pathological nodal stage (*p* = 0.22) and postoperative UICC stage (*p* = 0.222) showed no difference between the LR Group and the RR Group. Specimen resection quality scoring showed similar results between both groups (*p =* 0.355). A mercury score of 1 (good) was achieved in the majority of the resected specimens in both groups (LR Group: 86.67% vs. RR Group: 88.14%), and a score of 3 (bad) was observed only in two laparoscopic cases (3.33%). The number of retrieved lymph nodes did not differ between groups (LR Group: 22 (16–27) vs. 24 (15–30), *p* = 0.512). Oncological findings are shown in Table 3. 

## 4. Discussion

The introduction of a robotic-assisted surgery offers an alternative minimally invasive technique for malignant colorectal disease. In this study, we could clearly demonstrate that robotic surgery is a safe method regarding perioperative morbidity compared to conventional laparoscopic colorectal resections within the first three years after a robotic program was introduced. Moreover, oncological findings were similar compared to conventional laparoscopy. A lower conversion rate and a higher rate of intracorporeal anastomosis confirmed the advantages of the robotic platform. The strength of this study was the source of data, a prospective, specific database from two different hospitals with a highly experienced team of colorectal surgeons. Furthermore, similar baseline characteristics confirmed the assumption that no case selection bias occurred just to produce a better outcome. 

In the previous published literature, it is well-documented that robotic-assisted surgery has similar short-term findings compared to conventional laparoscopic techniques [18,19,20,21]. The goal of this study was to evaluate the feasibility and safety of the robotic platform regarding perioperative morbidity and histopathological findings compared to conventional laparoscopic techniques. In accordance with previous studies [9,18,19], we observed a longer operative time in the robotic-assisted group, which could be explained by the surgeons’ lack of experience using the system at the beginning of the robotic program. A detailed analysis of the duration of the operation confirms this trend: Figure 1 shows that operative time decreased with the number of procedures. 

However, trocar placement and docking of the robotic-assisted system also require time in the case of a completed learning curve [18]. The most relevant outcome of this study was the significantly lower conversion rate compared to conventional laparoscopic resection (LR Group: 16 (19.51%) vs. RR Group: 5 (5.38%), *p* = 0.004), which is consistent with the results of the previous published literature [9,18,22]. Several laparoscopic studies, including randomized trials (COST [23], CLASSIC [24]), demonstrated that conversions are associated with a worse oncological outcome and higher postoperative complication rates [19,25]. The robotic platform may allow surgeons to complete a difficult dissection, especially in male mid- and low-rectal-cancer patients. However, the conversion rate in the laparoscopic group was higher compared to that shown in the existent literature (LR Group: 19.51%). Most conversions occurred in laparoscopic right-sided resections (8/16 = 50%). On the one hand, this could be explained by the increased difficulty of right-sided oncological resections due to anatomical variability compared to left-sided resections, and on the other hand, by the well-documented advantage of robotic surgery in soft tissue dissection.

In contrast to previous published meta-analyses [18,26,27], our data do not show a significant difference regarding hospital stay, which could be explained by the fact that time to first flatus is in general faster in laparoscopy [3], and bowel obstruction is a rare complication after minimally invasive procedures, independently of either laparoscopic or robotic-assisted techniques. Another important outcome of this study should be interpreted with caution: significantly more defunctioning stomas were created in the RR group (LR Group: 16 (32%) vs. RR Group: 39 (55.7%), *p* = 0.006), which was also significant in the PSM. This could be explained by the higher rates of rectal resections in the robotic group. Before the introduction of the robotic system, only selected patients with mid and low rectal cancer underwent surgery laparoscopically. 

Postoperative morbidity, such as anastomotic leakage, bowel obstruction, postoperative bleeding or intra-abdominal abscess, did not differ between groups, which is in line with the previous reported literature [25]. The results further represent the safety and efficacy of both robotic-assisted and laparoscopic colorectal surgery with respect to short-term findings. 

Regarding histopathological findings, we could not observe any difference in terms of harvested lymph nodes and quality of resected specimens. Quality of resected specimens is reported to be improved with robotic surgery in some previous studies, especially in the case of low anterior resections [28,29]. It should be noted that the rate of grade 1 resections was high in both groups (LR Group: 86.67% vs. RR Group: 88.14%). Only in two laparoscopic cases was a poor score achieved. Quality of specimens and the number of harvested lymph nodes are essential for long-term oncological outcomes [25,30]. 

In right-sided colonic resections, we observed significantly more intracorporeal performed anastomosis in the robotic group (LR Group: 26.31% vs. RR Group: 76.92%, *p* = 0.008). The use of different types of anastomoses reflects the current clinical practice. Extracorporeal anastomosis is often reserved for laparoscopic resections, while intracorporeal anastomosis is performed in robotic-assisted procedures due to different levels of technical difficulty [18,26]. Intracorporeal anastomosis in robotic-assisted procedures seems to be easier to perform due to the advanced instrument triangulation, wristed instruments, better ergonomics and the efficient way of performing sutures. However, the higher rate of intracorporeal anastomosis in the RR Group does not prove the benefits of robotic surgery. Docking of the system requires operation time. Performing an extracorporeal anastomosis in robotic surgery would also require removing the robot from the patient, which might be time-consuming. This fact must be considered in the interpretation of the higher rate of intracorporeal anastomosis in the robotic group of this study. 

However, it is well-known that during the introduction of a robotic program, surgeons with extensive experience in minimally invasive colorectal surgery perform most of the cases. In addition, these are mostly patients with predictably better outcomes and fewer risk factors, such as a lower BMI, lower rate comorbidities and lower tumor stages. Both laparoscopic and robotic resections showed similar baseline characteristics as well as comparable pre- and postoperative UICC stages in this study. To reduce bias, we additionally performed a PSM. In this analysis, baseline characteristics as well as pre- and postoperative UICC stages did not differ. We observed in general a clear shift to pelvic surgery, where we see the biggest advantage of robotic-assisted surgery (mid- and low-rectal-cancer patients: LR Group: 52.5%, RR Group: 64.52%). Although this was not significant, other tumor locations than rectal cancer were similar or more often observed in the laparoscopic group. 

This study presented a few limitations to be mentioned: First, the study was of a retrospective design, which implicates a selection bias, even if not significant, in patients’ characteristics, as shown before. Second, we did not address cost in this study, because of different and very complicated clearing systems in our institutions. In addition, due to a relatively small sample size possible significant differences between both groups may not be detected. Third, in the study, all operations were performed in two different hospitals with some differences in standard operation procedures of each hospital. Even if the surgical procedure did not differ, pre- and postoperative applications (bowel preparation, oral food intake) varied. Finally, because of the retrospective design of this study, important variables (time of oral intake and pain scores) were lacking. 

In this study, all operations were performed by specialized surgeons. Surgical training and real clinical experience for surgical residents or surgeons with limited practical knowledge in colorectal surgery were lacking. Dual consoles would improve teaching possibilities. Without dual consoles, the introduction of residents or surgeons with limited experience into robotic colorectal surgery is not suitable because of limited possibilities to guide the procedures. 

Although both laparoscopic and robotic-assisted surgery are safe techniques for colorectal malignancies, robotic surgery may be more beneficial in mid- and low-rectal-cancer patients [31]. However, our data show a lower conversion rate of robotic-assisted colectomies compared to conventional laparoscopic resections. Oncological findings, the number of harvested lymph nodes and specimen scoring did not differ between groups. We could clearly demonstrate that the introduction of the robotic platform had no influence on short-term morbidity and oncological findings. 

## 5. Conclusions

In this study, we could clearly demonstrate robotic-assisted colorectal cancer surgery as feasible and safe regarding postoperative morbidity. Oncological findings and the number of harvested lymph nodes were comparable to conventional laparoscopy during the introduction of a robotic program. We can assume that with increasing expertise in robotic-assisted surgery, this positive difference will become even more pronounced.

## Figures and Tables

**Figure 1 cancers-14-03208-f001:**
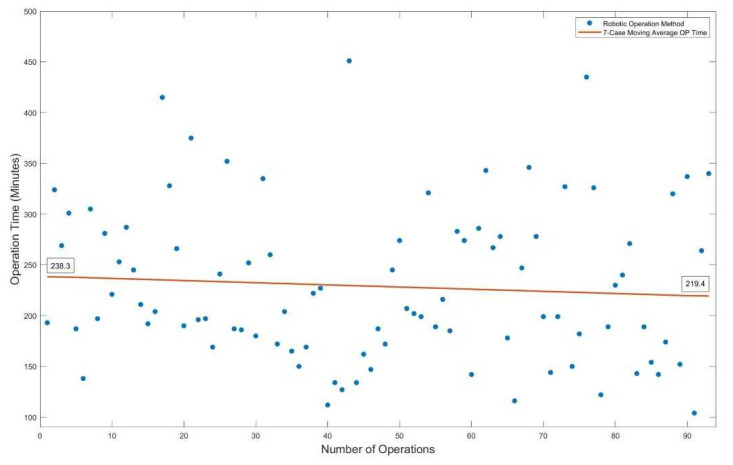
Operation time trend decreased with number of performed cases.

**Table 1 cancers-14-03208-t001:** Patients’ characteristics.

	Entire Cohort			Propensity-Score-Matched Cohort		
	LR Group 82	RR Group 93	*p*-Value	LR Group 63	RR Group 63	*p*-Value
Sex, male/female, *n* (%)	44 (53.66%)/38 (46.34%)	60 (64.52%)/33 (35.48%)	0.144	38 (60.32%)/25 (39.68%)	37 (58.73%)/26 (41.27%)	0.856
Age, median (IQR), y	64.0 (58.0–75.0)	68.0 (60.0–75.0)	0.438	65.0 (60.0–78.0)	70.0 (62.0–76.0)	0.519
BMI, median (IQR), kg/m^2^	25.4 (23.0–29.0)	25.1 (23.0–28.0)	0.654	25.0 (23.0–28.0)	24.7 (22.0–28.0)	0.583
ASA (average), *n* (%)			0.122			0.222
I	3 (3.66%)	12 (12.9%)		3 (4.76%)	9 (14.29%)	
II	39 (47.56%)	41 (44.09%)		29 (46.03%)	25 (39.68%)	
III	39 (47.56%)	40 (43.01%)		30 (47.62%)	29 (46.03%)	
IV	1 (1.22%)	0 (0.0%)		1 (1.59%)	0 (0.0%)	
V	0	0		0	0	
Tumor localization, *n* (%)						
Right	19 (23.75%)	13 (13.98%)	0.116	13 (21.31%)	10 (15.87%)	0.489
Left (+upper rectum)	19 (23.75%)	20 (21.51%)	0.792	14 (22.95%)	16 (25.4%)	0.676
Rectum (mid, low rectum)	42 (52.5%)	60 (64.52%)	0.075	34 (55.74%)	37 (58.73%)	0.473
CEA level preoperative, median (IQR), µg/L	2.4 (1.0–4.0)	2.1 (1.0–4.0)	0.468	2.6 (2.0–4.0)	2.2 (1.0–4.0)	0.425
Previous abdominal surgery, *n* (%)	18 (21.95%)	32 (34.41%)	0.056	13 (20.63%)	22 (34.92)	0.037
Comorbidities, *n* (%)			0.091			0.066
Yes	27 (32.93%)	41 (45.56%)		19 (30.16%)	28 (46.67%)	
No	55 (67.07%)	49 (54.44%)		44 (69.84%)	32 (53.33%)	
Comorbidities, %			0.328			0.351
Coronary disease	24.39%	13.98%		26.98%	14.29%	
Pulmonary insufficiency	4.88%	6.45%		6.35%	4.76%	
Obesity	8.54%	11.83%		4.76%	9.52%	
Kidney disease	9.76%	7.53%		4.76%	4.76%	
Hypertension	39.02%	37.63%		44.44%	33.33%	
Chronic renal failure	3.66%	2.15%		4.76%	1.59%	
Insult	4.88%	2.15%		6.35%	1.59%	
Atrial fibrillation	7.32%	4.3%		9.52%	4.76%	
Clinical stage (UICC), *n* (%)			0.611			0.459
I	18 (29.03%)	17 (23.94%)		6 (22.22%)	6 (42.86%)	
II	26 (41.94%)	25 (35.21%)		10 (37.04%)	4 (28.57%)	
III	17 (27.42%)	28 (39.44%)		11 (40.74%)	4 (28.57%)	
Preoperative therapy (rectal cancer), *n* (%)			0.344			0.07
Short-term (5 × 5Gy)	3 (3.66%)	2 (2.15%)		3 (4.76%)	1 (1.59%)	
Long-term (50.4 Gy + Chemotherapy)	18 (21.95%)	27 (29.03%)		15 (23.81%)	15 (23.81%)	

IQR = interquartile range, BMI = body mass index, ASA = American Society of Anesthesiologists, CEA = carcinoembryonic antigen, UICC = Union Internationale Contre le Cancer. Values are given as median and IQR (interquartile range) or numbers and percentage.

**Table 2 cancers-14-03208-t002:** Operative method, perioperative results and postoperative complications.

	Entire Cohort			Propensity-Score-Matched Cohort		
	LR Group 82	RR Group 93	*p*-Value	LR Group 63	RR Group 63	*p*-Value
Operative method, *n* (%)			0.14			0.397
Right	19 (23.17%)	13 (13.98%)		14 (22.22%)	10 (15.87%)	
Left (+upper rectum)	19 (23.75%)	20 (21.51%)		14 (22.95%)	16 (25.4%)	
Rectum (mid, low rectum)	42 (52.5%)	60 (64.52%)		33 (52.38%)	37 (58.73%)	
Transverse	2 (2.44%)	0 (0%)		2 (3.17%)	0 (0%)	
Anastomosis			<0.001			0.008
S-S	21 (25.61%)	13 (13.98%)		15 (28.3%)	11 (18.97%)	
E-E	31 (37.8%)	26 (27.96%)		24 (45.28%)	16 (27.59%)	
S-E	16 (19.51%)	46 (49.46%)		12 (22.64%)	30 (51.72%)	
Colo-anal	2 (2.44%)	2 (2.15%)		2 (3.77%)	1 (1.72%)	
Intracorporeal/extracorporeal (right hemicolectomy), *n* (%)			0.008			0.11
extracorporeal	14 (73.68%)	3 (23.07%)		10 (15.87%)	3 (4.76%)	
intracorporeal	5 (26.31%)	10 (76.92%)		3 (4.76%)	6 (9.52%)	
Protective defunctioning stoma, *n* (%)	16 (32%)	39 (55.7%)	0.006	20 (31.7%)	37 (58.73%)	0.03
Operation time, median (IQR), min	200.0 (150.0–243.0)	204.0 (174.0–278.0)	0.045	205.0 (154.0–244.0)	193.0 (158.0–252.0)	0.915
Time to first flatus, median (IQR), d	2.0 (1.0–2.0)	2.0 (1.0–2.0)	0.803	2.0 (1.0–2.0)	2.0 (1.0–2.0)	0.768
Hb preoperative, median (IQR), g/l	13.4 (12.0–15.0)	13.2 (12.0–14.0)	0.493	13.3 (12.0–14.0)	13.3 (12.0–15.0)	0.757
Hb postoperative, median (IQR), g/l	11.65 (10.0–13.0)	11.3 (10.0–12.0)	0.242	11.8 (10.0–13.0)	11.3 (10.0–12.0)	0.468
Complications, *n* (%)	18.29%	21.51%	0.596	19.05%	15.87%	0.639
Anastomotic leakage	4 (4.88%)	4 (4.3%)		3 (4.76%)	2 (3.17%)	
Wound infection	4 (4.88%)	3 (3.23%)		3 (4.76%)	2 (3.17%)	
Bleeding	1 (1.22%)	2 (2.15%)		1 (1.59%)	0 (0.0%)	
Intra-abdominal abscess/infection	1 (1.22%)	2 (2.15%)		1 (1.59%)	1 (1.59%)	
Bowel obstruction	0 (0%)	0 (0%)		0 (0%)	0 (0%)	
Renal dysfunction	0 (0%)	1 (1.08%)		0 (0%)	1 (1.59%)	
Others *	5 (6.1%)	8 (8.6%)		4 (6.35%)	4 (6.35%)	
Clavien–Dindo Classification, *n* (%)			0.855			0.912
I	1 (1.22%)	3 (3.23%)		1 (1.59%)	1 (1.59%)	
II	7 (8.54%)	8 (8.6%)		5 (7.94%)	4 (6.35%)	
III	6 (7.32%)	7 (7.53%)		5 (7.94%)	3 (4.76%)	
IV	0 (0%)	0 (0%)		0 (0%)	0 (0%)	
V	1 (1.22%)	2 (2.15%)		1 (1.59%)	2 (3.17%)	
Conversion to open procedure, *n* (%)	16 (19.51%)	5 (5.38%)	0.004	13 (20.63%)	3 (4.76%)	0.006
Stoma reversal time, median (IQR), d	75.5 (65.0–394.0)	78.0 (31.0–110.0)	0.339	220.5 (73.0–464.0)	70.0 (22.0–83.0)	0.068
Duration of hospital stay, median (IQR)	9.0 (7.0–14.0)	9.0 (6.0–13.0)	0.928	9.0 (7.0–14.0)	9.0 (6.0–12.0)	0.149

IQR = interquartile range, Hb = hemoglobin, S-S = side to side, E-E = end to end, S-E = side to end. Values are given as median and IQR (interquartile range) or numbers and percentage. * Including urinary infection/retention, pulmonary and cardiac complications.

**Table 3 cancers-14-03208-t003:** Histopathological findings, specimen score and number of retrieved lymph nodes.

	Entire Cohort			Propensity-Score-Matched Cohort		
	LR Group 82	RR Group 93	*p*-Value	LR Group 63	RR Group 63	*p*-Value
Pathological T Stage, *n* (%)			0.834			0.896
Tis	2 (2.47%)	6 (6.45%)		1 (1.59%)	2 (3.17%)	
T1	15 (18.52%)	16 (18.18%)		12 (19.05%)	12 (19.05%)	
T2	29 (35.36%)	27 (30.68%)		24 (38.1%)	20 (31.75%)	
T3	32 (39.51%)	40 (43.01%)		24 (38.1%)	25 (39.68%)	
T4	4 (4.94%)	4 (4.55%)		2 (3.17%)	3 (4.76%)	
Pathological N Stage, *n* (%)			0.22			0.847
N0	60 (73.17%)	60 (64.51%)		44 (69.84%)	43 (68.25%)	
N+	22 (27.16%)	33 (35.87%)		19 (30.16%)	20 (31.75%)	
Postoperative UICC Stage, *n* (%)			0.225			0.295
0	3 (3.66%)	6 (6.45%)		1 (1.59%)	2 (3.17%)	
I	34 (41.46%)	33 (35.87%)		27 (42.86%)	26 (41.27%)	
II	20 (24.39%)	18 (19.57%)		15 (23.81%)	12 (19.05%)	
III	25 (30.49%)	31 (33.7%)		20 (31.75%)	20 (31.75%)	
IV	0 (0%)	5 (5.43%)		0 (0%)	3 (4.76%)	
Mercury Score, *n* (%)			0.355			0.343
I	52 (86.67%)	52 (88.14%)		39 (84.78%)	34 (85.0%)	
II	6 (10.0%)	7 (11.86%)		5 (10.87%)	6 (15.0%)	
III	2 (3.33%)	0 (0.0%)		2 (4.35%)	0 (0.0%)	
Number of retrieved lymph nodes, median (IQR)	22.0 (16.0–27.0)	24.0 (15.0–30.0)	0.512	22.0 (16.0–26.0)	23.0 (15.0–28.0)	0.942

IQR = interquartile range, UICC = Union Internationale Contre le Cancer. Values are given as median and IQR (interquartile range) or numbers and percentage.

## Data Availability

The datasets generated during and/or analyzed during the current study are available from the corresponding author on reasonable request.

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
