# Peer review of "Is Robotic Assisted Colorectal Cancer Surgery Equivalent Compared to Laparoscopic Procedures during the Introduction of a Robotic Program? A Propensity-Score Matched Analysis"

_cancers, 2022, doi:10.3390/cancers14133208_

Round 1

Reviewer 1 Report

Authors aimed to compare short-term outcome and oncological findings of robotic assisted colorectal resections with those of conventional laparoscopic surgery within the first three years after the introduction of the robotic platform from two centres. Their results indicated that laparoscopic resection (LR Group) was performed in 82 cases and 93 patients were treated robotic assisted (RR Group). Patients’ characteristics did not differ between both groups. In right sided resections an intracorporeal anastomosis was significantly more often performed in the RR Group (p=0.008). Operative time showed to be significantly shorter in the LR Group (p=0.045). Conversions to open surgery did occur more often in the LR Group (p=0.004). Postoperative morbidity, number of harvested lymph nodes, quality of resection and postoperative tumor stage did not differ between both groups. Conclusion: Overall, they concluded that they could clearly demonstrate robotic assisted colorectal cancer surgery as effective, feasible and safe regarding postoperative morbidity and oncological findings compared to conventional laparoscopy during the introduction of a robotic system. The results seems informative and interesting; however, there are a lot of criticisms and have several issues that the authors need to address before the manuscript is suitable for publication.

Major Compulsory Revisions:        

1.     In Introduction section: To our knowledge, there are no clear data existing, how robotic assisted surgery is influencing the surgical outcome compared to other minimal invasive colorectal resections at the beginning of a robotic program. However, too many similar studies have the same study results even no relevant in their article title. For example, 1. Robotic Versus Laparoscopic Surgery for Colorectal Cancer: a Case-control Study. Radiol Oncol 2021; 55(4): 433–438; 2. Effective implementation and adaptation of structured robotic colorectal programme in a busy tertiary unit. J Robot Surg 2021;15(5):731-739; 3. Robotic versus laparoscopic rectal resection for sphincter-saving surgery: pathological and short-term outcomes in a single-center analysis of 130 consecutive patients. Surg Endosc 2017 Oct;31(10):4085-4091, etc. Therefore, the well-known information is lack of interests for readers.

2.     Another major concern was that only 82 cases in LR group and 93 patients in RR group were enrolled, and a propensity score matching method is suggested for the adjustment of bias. Particularly, in Methods section, authors have mentioned that the secondary outcomes were postoperative morbidity and oncological staging as a possible selection bias for robotic surgery. Moreover, if all 82 cases in LR group and 93 patients in RR group were consecutively enrolled?   

3.     Table 2. Operative method, perioperative results and postoperative complications. Apparently, the higher percentage of protective defunctional stoma in RR group might result from the higher percentage of rectal cancer patients in RR group. Therefore, it was not feasible to have this variable in the comparison of all CRC patients. A lot of important variables of surgical outcomes including if faster recovery, less pain, and earlier oral intake were lack in the current study. Regarding post-operative complications, urinary retention or infection and pulmonary complication need to be addressed. And intra-abdominal abscess should be extended to be intra-abdominal infection/abscess.   

4.     Figure 1. Operation time trend is decreasing with number of performed cases. The 7-case simple moving average method was suggested to analyze the learning curve of console time and operation time. Furthermore, the first plateau of patient’s number of console time and operation time was needed to be indicated.

5.     In Discussion section: This study presents a few limitations to be mentioned. Second, the results observed in this study, especially the low conversion rate to an open procedure could be explained due to the high level of experience from surgeons that performed this minimal invasive colorectal surgery. Conversion to open surgery rate in LR group and RR group was 19.51% and 5.38%, respectively. In fact, their conversion to open surgery rate was not significantly impressive in the comparison with other well-experienced surgeons.

6.     In conclusion section: Moreover, our data underlined the advantage of robotic assisted surgery especially in the pelvis and in performing intracorporeal anastomosis in right-sided resections. However, from the limited results from the current study that authors could not have the above conclusions.  

Minor Essential Revisions: 

1.     Statistical analysis was realizing using the SPSS (IBM, New York). Please indicate the version used. 

Author Response

Dear Reviewer!

Thank you very much for reviewing this manuscript and for all your recommendations. To reduce bias reviewer 1 suggested a propensity score matched analysis. Therefore, we renamed the title. Because of this we carefully reviewed the manuscript and changed the statistical analysis. All corrections are marked and highlighted. I hope we answered all your queries correctly to improve this manuscript. Please don’t hesitate to contact me if you have further questions/suggestions.

Kind regard

  1. Tschann

Point for point answer:

  1. In Introduction section: To our knowledge, there are no clear data existing, how robotic assisted surgery is influencing the surgical outcome compared to other minimal invasive colorectal resections at the beginning of a robotic program. However, too many similar studies have the same study results even no relevant in their article title. For example, 1. Robotic Versus Laparoscopic Surgery for Colorectal Cancer: a Case-control Study. Radiol Oncol 2021; 55(4): 433–438; 2. Effective implementation and adaptation of structured robotic colorectal programme in a busy tertiary unit. J Robot Surg 2021;15(5):731-739; 3. Robotic versus laparoscopic rectal resection for sphincter-saving surgery: pathological and short-term outcomes in a single-center analysis of 130 consecutive patients. Surg Endosc 2017 Oct;31(10):4085-4091, etc. Therefore, the well-known information is lack of interests for readers.

Thank you very much for this recommendation: We tried to introduce this article and cited some introductions from studies who compared laparoscopic with robotic surgery. Therefore, we added the ROLLARR Trial at the beginning of the introduction as citation specially to highlight the advantages of robotic surgery (P3, L64, 71-74).  

  1. Another major concern was that only 82 cases in LR group and 93 patients in RR group were enrolled, and a propensity score matching method is suggested for the adjustment of bias. Particularly, in Methods section, authors have mentioned that the secondary outcomes were postoperative morbidity and oncological staging as a possible selection bias for robotic surgery. Moreover, if all 82 cases in LR group and 93 patients in RR group were consecutively enrolled?   

You are right. For that reason, we performed a propensity score matched analysis. We added this in the abstract, method- and result section and in Table 1, 2 and 3 (P1, P5-8). Additionally, we added a sentence that all patients were consecutively enrolled P3 L88). Therefore, title was renamed.

  1. Table 2. Operative method, perioperative results and postoperative complications. Apparently, the higher percentage of protective ifunctional stoma in RR group might result from the higher percentage of rectal cancer patients in RR group. Therefore, it was not feasible to have this variable in the comparison of all CRC patients. A lot of important variables of surgical outcomes including if faster recovery, less pain, and earlier oral intake were lack in the current study. Regarding post-operative complications, urinary retention or infection and pulmonary complication need to be addressed. And intra-abdominal abscess should be extended to be intra-abdominal infection/abscess.

You are right. We critically discussed this fact and added this into the discussion section (P10, L251-255). But we did not compare this variable with all colorectal resections. Percentage is related to rectal resections only. Regarding faster recovery, less pain, and earlier oral intake: you are right, but the study design was retrospective, therefore we have no data. We added this in the limitation section (P11, L294). Regarding urinary retention/infection and pulmonary complications: we are sorry, but these complications were included as “others”. We added a “*” in table legend (P8, L206). 

  1. Figure 1. Operation time trend is decreasing with number of performed cases. The 7-case simple moving average method was suggested to analyze the learning curve of console time and operation time. Furthermore, the first plateau of patient’s number of console time and operation time was needed to be indicated.

A new Figure was added as Figure 1 and described in the methods section (P5, L159-160)

  1. In Discussion section: This study presents a few limitations to be mentioned. Second, the results observed in this study, especially the low conversion rate to an open procedure could be explained due to the high level of experience from surgeons that performed this minimal invasive colorectal surgery. Conversion to open surgery rate in LR group and RR group was 19.51% and 5.38%, respectively. In fact, their conversion to open surgery rate was not significantly impressive in the comparison with other well-experienced surgeons.

You are right. We deleted this sentence (P10, L291). Furthermore, we added a critical view on conversion rate into the discussion section (P10, L251-255).

  1. In conclusion section: Moreover, our data underlined the advantage of robotic assisted surgery especially in the pelvis and in performing intracorporeal anastomosis in right-sided resections. However, from the limited results from the current study that authors could not have the above conclusions.

You are right. We deleted this sentence and highlighted the comparable oncological findings (P11, L304)

Reviewer 2 Report

The authors compared the traditional laparoscopic technique (LT) with the robotic assisted technique (RA) for colorectal cancer in the first three years after implementation of the robotic program regarding short-term outcome and oncological quality during the learning curve. They only found significant differences in the conversion rate (lower in RA) and the length of procedure (slightly shorter in LT).

This is an interesting manuscript dealing with a very interesting topic. But there are some required corrections, additions and also a few remaining questions, that should be clarified and discussed, respectively:

Methods:

When were the procedures performed in the particular centers (time period)?

Was the robotic platform established in both centers at the same time?

Were operations taken into account from the first robotic procedure?

How are “specialized colorectal surgeons with a high experience in both open and laparoscopic surgery” defined?

How was the choice made on the particular approach? Did it depend on the respective surgeon or the time period?

2.2. Histopathological examination: 

“scoring about the quality 139 of the resection” is listed twice.

Results:

What is the distribution of procedures between the two centers? Is this a comparison of two centers?

How many surgeons were involved in this study?

How often was single incision- or reduced port surgery performed?

Could you please divide the rectal resections into the thirds (upper, mid, low rectum)?

Three patients died within the postoperative course due to severe complication. Please provide more information.

There were more patients with a Mercury Score than rectal cancers. Please clarify this.

The sum of the pathological stages as well as pathological N stages and postoperative UICC stages does not correspond to the total number of patients. Please clarify this.

Discussion:

Why was oral antibiotic prophylaxis only delivered at the surgical department of Salzburg?

Conversion rate to open procedure seems to be high, even in robotic assisted technique, particularly for high experienced surgeons. Almost one in five laparoscopic patients was converted (19.5%)! Please discuss this point.

Significantly more rectal cancers were operated on robotically than traditionally laparoscopically, although this difference was not significant. Please discuss this point.

“… a higher rate of intracorporeal anastomosis confirmed the advantages of the robotic platform.” This does not depend on the choice of the procedure. You can also perform an intracorporeal anastomosis laparoscopically. The fact that you do not do this so often does not prove the benefit of RA.

A trend toward the robotic assisted technique in patients with Clinical stage (UICC) III can be seen in this study, although this difference was not significant. Please discuss this point.

The results of figure 1 should be shown in the results section and previously explained in the methods section.

References:

Critical are the self-Citations 3 (does not show the rapid increase of laparoscopic colectomies) as well as 7 (non-comparable analysis).

Conclusion:

I do not think that the “data underlined the advantage of robotic assisted surgery … in performing intracorporeal anastomosis in right-sided resections”.

Author Response

Dear Reviewer!

Thank you very much for reviewing this manuscript and for all your recommendations. To reduce bias reviewer 1 suggested a propensity score matched analysis. Therefore, we renamed the title. Because of this we carefully reviewed the manuscript and changed the statistical analysis. All corrections are marked and highlighted. I hope we answered all your queries correctly to improve this manuscript. Please don’t hesitate to contact me if you have further questions/suggestions.

Kind regard

  1. Tschann

Point for point answer:

  1. Statistical analysis was realizing using the SPSS (IBM, New York). Please indicate the version used. 

You are right. Corrected (P5, L155).

The authors compared the traditional laparoscopic technique (LT) with the robotic assisted technique (RA) for colorectal cancer in the first three years after implementation of the robotic program regarding short-term outcome and oncological quality during the learning curve. They only found significant differences in the conversion rate (lower in RA) and the length of procedure (slightly shorter in LT).

This is an interesting manuscript dealing with a very interesting topic. But there are some required corrections, additions and also a few remaining questions, that should be clarified and discussed, respectively:

Methods:

  1. When were the procedures performed in the particular centers (time period)?

 Added in the methods section (P3, L86, 87).

  1. Was the robotic platform established in both centers at the same time?

No. In the Hospital in Feldkirch the robotic platform was introduced in 2019, and in Salzburg one year earlier. Added into the methods section (P3, L86-87)

  1. Were operations taken into account from the first robotic procedure?

Yes. All consecutive colorectal resections from the beginning of the robotic platform (including the first resections in both centres). Highlighted in P3, L85.

  1. How are “specialized colorectal surgeons with a high experience in both open and laparoscopic surgery” defined?

All surgeons who performed the resections had at least 10 years of experience in colorectal surgery. We added this to the methods section (P3, L87).

  1. How was the choice made on the particular approach? Did it depend on the respective surgeon or the time period?

In both centers the robotic platform is also used by urologists and gynecologists. Therefore, the robotic system was not available all the time. The indication was independent to surgeons’ preferences. 

  1. Histopathological examination: 

“scoring about the quality 139 of the resection” is listed twice.

Thank you. We deleted one sentence in the methods section (P5, L144).

Results:

  1. What is the distribution of procedures between the two centers? Is this a comparison of two centers?

We added the number of performed cases of each hospital in the methods section (P3, L86, 87). It was not a comparison between the hospitals. The patients’ characteristics and perioperative findings did not differ between both hospitals. Moreover, we didn’t intend to compare to hospitals against each other.

  1. How many surgeons were involved in this study?

4 surgeons, respectively.

  1. How often was single incision- or reduced port surgery performed?

We only recorded laparoscopic or robotic technique independent of SILS or SIL+1. At the Hospital in Feldkirch the conventional laparoscopic resections were performed in reduced port technique (SIL+1).

  1. Could you please divide the rectal resections into the thirds (upper, mid, low rectum)?

You are right. This was corrected in table 2.

  1. Three patients died within the postoperative course due to severe complication. Please provide more information.

 A specification was added into the result section (P7, L198, 199).

  1. There were more patients with a Mercury Score than rectal cancers. Please clarify this.

In all rectal cancer cases a Mercury Score was routinely performed. In all other cases it was performed on special request, this fact was additionally described in the methods section (P5, L144).

  1. The sum of the pathological stages as well as pathological N stages and postoperative UICC stages does not correspond to the total number of patients. Please clarify this.

 You are right. We reviewed the data and corrected table 3. The p-values didn’t change.

Discussion:

  1. Why was oral antibiotic prophylaxis only delivered at the surgical department of Salzburg?

For sure, this is one limitation. Even if surgical procedure and quality was good in both hospitals, pre- and postoperative SOP’s differed. We added this fact into limitations (P11, L291, 292).

  1. Conversion rate to open procedure seems to be high, even in robotic assisted technique, particularly for high experienced surgeons. Almost one in five laparoscopic patients was converted (19.5%)! Please discuss this point.

You are right. The rate is higher compared to existent literature. We tried to clarify in the discussion section. The half of the conversions occurred in right sided resections. This could be explained on the one hand by the anatomical variability, on the other hand by the advantages of robotic surgery in soft tissue dissection (P11, L242-246).

  1. Significantly more rectal cancers were operated on robotically than traditionally laparoscopically, although this difference was not significant. Please discuss this point.

 You are right, we added a sentence into the discussion section (P11, L284, 285).

  1. “… a higher rate of intracorporeal anastomosis confirmed the advantages of the robotic platform.” This does not depend on the choice of the procedure. You can also perform an intracorporeal anastomosis laparoscopically. The fact that you do not do this so often does not prove the benefit of RA.

This is right. We added a critical view on this point (P11, L271-276)

  1. A trend toward the robotic assisted technique in patients with Clinical stage (UICC) III can be seen in this study, although this difference was not significant. Please discuss this point.

To reduce selection bias we additionally performed a propensity score match analysis. In this we observed similar rates of UICC stages. We added a sentence in the discussion section (P11, L281, 282).

  1. The results of figure 1 should be shown in the results section and previously explained in the methods section.

The figure was added into the result section and additionally explained in the methods section (P7).

References:

  1. Critical are the self-Citations 3 (does not show the rapid increase of laparoscopic colectomies) as well as 7 (non-comparable analysis).

We carefully reviewed the references: We deleted one self-citation.

Conclusion:

  1. I do not think that the “data underlined the advantage of robotic assisted surgery … in performing intracorporeal anastomosis in right-sided resections”.

You are right. We deleted this sentence.

Round 2

Reviewer 1 Report

There were some minor points to be elucidated. 

Please correct the following points:

1. Conversions to an open procedure did significantly more often occur in the LR Group 195 (LR Group: 16 [19.51%] vs. RR Group: 5 [5.38%], p=0.004), which was significant in the PSM analysis too (LR 196 Group: 13 [20.63%] vs. RR Group: 3 (4.76%), p= 0.006). Two group not too group.

2. Abbreviation :  Anastomosis: S-S, E-E, S-E should be present the full name in the footnote of Table 2.

3. Another important outcome of this study should be interpreted 251 with caution: Significantly more defunctioning stomas were created in the RR group (LR Group: 16 [32%] 252 vs. RR Group: 39 [55.7%], p=0.006), which was also significant in the PSM. This could be explained by the 253 higher rates of rectal resections in the robotic group. Before the introduction of the robotic system only 254 selected patients with mid and low rectal cancer underwent surgery laparoscopically. However, 33 (52.38%) vs 37 (58.73%) in mid/lower rectum in LR group vs RR group but 20 (50%) vs 37 (86.04%) in protective defunction stoma in LR group vs RR group. The considerable different numbers between mid/lower rectum and protective defunction stoma should be discussed in more details.

4. How about the robotic colorectal surgery for laparoscopic surgeon with limited experiences would be an interesting issue for robotic assisted colorectal cancer surgery during the introduction of a robotic program?

Author Response

Dear Reviewer!

Thank you very much for reviewing the revised manuscript and for all your recommendations. All corrections are marked and highlighted (green). I hope we answered all your queries correctly to improve this manuscript. Please don’t hesitate to contact me if you have further questions/suggestions.

Kind regard

  1. Tschann

Point for point answer:

  1. Conversions to an open procedure did significantly more often occur in the LR Group 195 (LR Group: 16 [19.51%] vs. RR Group: 5 [5.38%], p=0.004), which was significant in the PSM analysis too (LR 196 Group: 13 [20.63%] vs. RR Group: 3 (4.76%), p= 0.006). Two group not too group.

Corrected (P7, L200)

  1. Abbreviation :  Anastomosis: S-S, E-E, S-E should be present the full name in the footnote of Table 2.

Added to abbreviations in Table 2.

  1. Another important outcome of this study should be interpreted 251 with caution: Significantly more defunctioning stomas were created in the RR group (LR Group: 16 [32%] 252 vs. RR Group: 39 [55.7%], p=0.006), which was also significant in the PSM. This could be explained by the 253 higher rates of rectal resections in the robotic group. Before the introduction of the robotic system only 254 selected patients with mid and low rectal cancer underwent surgery laparoscopically. However, 33 (52.38%) vs 37 (58.73%) in mid/lower rectum in LR group vs RR group but 20 (50%) vs 37 (86.04%) in protective defunction stoma in LR group vs RR group. The considerable different numbers between mid/lower rectum and protective defunction stoma should be discussed in more details.

I’m sorry, this was a mistake. I corrected table 2 (highlighted in green). N of PSM did not change, but the percentage was wrong. Therefore, there is no difference between PSM and entire cohort.

Protective defunctional stoma, n (%)

16 (32%)

39 (55.7%)

0.006

20 (31.7%)

37 (58.73%)

0.03

  1. How about the robotic colorectal surgery for laparoscopic surgeon with limited experiences would be an interesting issue for robotic assisted colorectal cancer surgery during the introduction of a robotic program?

This is a very interesting point. We added a sentence into the discussion section (P11, L299-303)

Reviewer 2 Report

Thank you for revising your manuscript. All but three points are revised to my complete satisfaction. The three points are the following:

3. Was the robotic platform established in both centers at the same time?

No. In the Hospital in Feldkirch the robotic platform was introduced in 2019, and in Salzburg one year earlier. Added into the methods section (P3, L86-87)

- Please state this explicitly in the text.

9. How many surgeons were involved in this study?

4 surgeons, respectively.

- Please mention this in the manuscript.

10. How often was single incision- or reduced port surgery performed?

We only recorded laparoscopic or robotic technique independent of SILS or SIL+1. At the Hospital in Feldkirch the conventional laparoscopic resections were performed in reduced port technique (SIL+1).

- Please mention this in the manuscript.

Author Response

Dear Reviewer!

Thank you very much for reviewing the revised manuscript and for all your recommendations. All corrections are marked and highlighted (green). I hope we answered all your queries correctly to improve this manuscript. Please don’t hesitate to contact me if you have further questions/suggestions.

Kind regard

  1. Tschann

Point for point answer:

  1. Was the robotic platform established in both centers at the same time?

No. In the Hospital in Feldkirch the robotic platform was introduced in 2019, and in Salzburg one year earlier. Added into the methods section (P3, L86-87)

- Please state this explicitly in the text.

 Added into the text (P3, L88).

  1. How many surgeons were involved in this study?

4 surgeons, respectively.

- Please mention this in the manuscript.

 Added into the text (P3, L89)

  1. How often was single incision- or reduced port surgery performed?

We only recorded laparoscopic or robotic technique independent of SILS or SIL+1. At the Hospital in Feldkirch the conventional laparoscopic resections were performed in reduced port technique (SIL+1).

- Please mention this in the manuscript.

Added into the manuscript (P5, L141-143).